# Neuroprotection and Non-Invasive Brain Stimulation: Facts or Fiction?

**DOI:** 10.3390/ijms232213775

**Published:** 2022-11-09

**Authors:** Matteo Guidetti, Alessandro Bertini, Francesco Pirone, Gessica Sala, Paola Signorelli, Carlo Ferrarese, Alberto Priori, Tommaso Bocci

**Affiliations:** 1“Aldo Ravelli” Center for Neurotechnology and Experimental Brain Therapeutics, Department of Health Sciences, University of Milan, Via Antonio di Rudinì 8, 20142 Milan, Italy; 2Department of Electronics, Information and Bioengineering, Politecnico di Milano, Piazza Leonardo da Vinci, 32, 20133 Milan, Italy; 3Clinical Neurology Unit, “Azienda Socio-Sanitaria Territoriale Santi Paolo E Carlo”, Department of Health Sciences, University of Milan, Via Antonio di Rudinì 8, 20142 Milan, Italy; 4School of Medicine and Surgery, Milan Center for Neuroscience (NeuroMI), University of Milano-Bicocca, Monza, 20126 Milan, Italy; 5Biochemistry and Molecular Biology Laboratory, Department of Health Sciences, University of Milan, 20142 Milan, Italy; 6Department of Neurology, ASST-Monza, San Gerardo Hospital, Monza, 20126 Milan, Italy

**Keywords:** non-invasive brain stimulation, tDCS, rTMS, neuroprotection, Parkinson’s Disease, Alzheimer’s Disease, neurodegenerative disorders, pathological proteins, deep brain stimulation

## Abstract

Non-Invasive Brain Stimulation (NIBS) techniques, such as transcranial Direct Current Stimulation (tDCS) and repetitive Magnetic Transcranial Stimulation (rTMS), are well-known non-pharmacological approaches to improve both motor and non-motor symptoms in patients with neurodegenerative disorders. Their use is of particular interest especially for the treatment of cognitive impairment in Alzheimer’s Disease (AD), as well as axial disturbances in Parkinson’s (PD), where conventional pharmacological therapies show very mild and short-lasting effects. However, their ability to interfere with disease progression over time is not well understood; recent evidence suggests that NIBS may have a neuroprotective effect, thus slowing disease progression and modulating the aggregation state of pathological proteins. In this narrative review, we gather current knowledge about neuroprotection and NIBS in neurodegenerative diseases (i.e., PD and AD), just mentioning the few results related to stroke. As further matter of debate, we discuss similarities and differences with Deep Brain Stimulation (DBS)—induced neuroprotective effects, and highlight possible future directions for ongoing clinical studies.

## 1. Introduction

Non-Invasive Brain Stimulation (NIBS) techniques, including transcranial Direct Current Stimulation (tDCS), transcranial Alternating Current Stimulation (tACS) and repetitive Transcranial Magnetic Stimulation (rTMS), have been proposed for years to improve both motor and non-motor symptoms in a number of neurological conditions, comprising neurodegenerative disorders as Alzheimer’s (AD) and Parkinson’s Disease (PD) [1,2,3,4,5]. They are safe and promising tools for the modulation of cortical and, probably, sub-cortical activities [6]. A growing body of literature strengthens their use for the treatment of both speech disturbances and axial symptoms of PD (bradykinesia, falls and dysphagia), where conventional pharmacological approaches did not provide long-lasting changes over time [7]. Moreover, they proved a significant effect for the treatment of the so-called “freezing of gait” (FOG), which still remains a challenge for clinicians and neuroscientists [8,9,10,11]. However, there is a substantial lack of papers discussing their putative role as disease-modifying, neuroprotective therapies; this is of key importance because pharmacological treatments show merely a “symptomatic” effect, without any significant interference with disease progression over time [12]. In this review, we encompass current knowledge about NIBS and neuroprotection, discussing novel data and old concepts, both in animal and human models, and highlighting the possible use of these techniques in early phases of the neurodegenerative process. Moreover, we suggest a new view of tDCS and rTMS mechanisms of action, not based on either polarity or frequency-dependence of their after-effects, but on their ability to interfere with pathological protein accumulation and degradation in experimental models of neurodegenerative disorders. As a matter of debate and to give a more complete picture about neurostimulation and neuroprotection, we briefly provide a comparison between NIBS and invasive brain stimulation, as DBS (“Deep Brain Stimulation”), towards the goal of neuroprotection, both in degenerative and non-degenerative disorders; accordingly, DBS has recently demonstrated interesting results in animal models of schizophrenia and depression [13,14].

## 2. NIBS and Neuroprotection in Parkinson’s Disease

### 2.1. tDCS

Parkinson’s disease (PD) is the second most common neurodegenerative disorder affecting about 1% of the population >60 years of age [15]. The pathological hallmark of PD is the progressive degeneration of dopaminergic neurons in the substantia nigra and striatum [16], ultimately leading to motor (e.g., tremor, akinesia, rigidity, gait impairments) and non-motor (e.g., anxiety, depression, cognitive deficits) symptoms. However, the underpinning mechanisms remain unclear [17]. Multiple factors may contribute to neuronal damage, both in mutation related and in idiopathic forms of the disease [18], including biochemical factors, causing cellular stress accumulation, due to inflammation, oxidative stress and excitotoxicity, and leading to mitochondrial dysfunction, energy production loss and cell demise (e.g., mitochondrial dysfunction, defective protein degradation, neuroinflammation, oxidative stress, excitotoxicity), all of which are tightly linked to each other [19,20,21]. Available treatments are only symptomatic, and pharmacological therapies lead to several and disabling side effects over time [12].

Several neuromodulation techniques have been suggested, such as complementary treatment approach, both invasive [22,23,24] and non-invasive [25,26]. Among these, transcranial direct current stimulation (tDCS) showed a convincing therapeutic potential, with benefits both on motor and cognitive performances [27,28,29,30,31,32,33]. However, the mechanisms by which tDCS exerts its effects in PD patients are not fully understood, particularly at cellular and molecular levels [34]. Current knowledge suggests that tDCS increases the release of dopamine [35,36,37], modulates alpha-synuclein aggregation and autophagic degradation [34], alters neurotransmitters concentration (e.g., GABA, serotonin, glutamate) [38] and induces anti-apoptotic and anti-inflammatory effects [20,39]. However, these cellular effects have been collected either in vitro or in animal models, but not yet confirmed in human studies. In this scenario, tDCS is likely to enhance the expression of brain-derived neurotrophic factor (BDNF) [40], a neurotransmitter modulator and neurotrophic factor that supports neurogenesis [41,42] and survival of neurons [43]. Therefore, these findings suggest a possible neuroprotective effect of tDCS, which appears to be partially effective in restoring some of the biochemical defects associated with neurodegenerative diseases, as confirmed by animal studies [17,19,44,45] (see Figure 1). Indeed, current knowledge comes from neurotoxin-treated animal models, and shows preliminary and promising results in terms of tDCS-induced antioxidant function and survival of dopaminergic cells from neurotoxin-induced cell death [17,19,44,45].

In PD patients, the alteration of tyrosine hydroxylase activity (TH—an enzyme catalysing the precursor of dopamine, L-DOPA, in dopamine) reduces dopamine (DA) levels [46]. Besides, oxidative stress is increased and antioxidative processes are inhibited [19]. tDCS may have a role in neuronal protection acting on the response against oxidative stress, as suggested by Lee et al., 2019 [17] and Li et al., 2015 [19] (see Table 1). Anodal tDCS applied on mice preserves dopaminergic neurons after the injection of the neurotoxin 1-methyl-4-phenyl-1,2,3,6-tetrahydropyridine (MPTP) [17] and increases both DA and TH content [19]. Also, it reduces the decrease of antioxidant enzymes activities (superoxide dismutase, SOD; and glutathione peroxidase, GSH-Px) induced by MPTP, ultimately improving the survival response in the nigral-striatal area. However, antioxidative results might not be a direct effect of such a response and may not be driven directly by the stimulation, but rather a consequence of enhanced secretion of BDNF induced by tDCS, as shown by previous studies [38].

The major target of oxidative stress is the mitochondrion-increased production (or decreased capacity to eliminate) of free radicals, known to induce neuronal death via disruption of mitochondrial function [47]. Several findings suggest a pivotal role of mitochondrial dysfunctions in PD pathogenesis [48]. Lee et al., 2019 [17] demonstrated in mice that anodal tDCS exerts a neuroprotective effect against MPTP toxicity, by normalizing mitophagy activation, enhancing mitochondrial biogenesis and restoring mitochondrial damage (see Table 1) [17,49]. tDCS also decreases the effects of MPTP, i.e., increased expression of mitophagy-related proteins (PTEN-induced putative kinase 1, PINK1; Parkin; and microtubule-associated protein light chain 3, LC3), PINK1/Parkin upregulation and enhanced autophagic flux [17]. Mitochondrial biogenesis-related proteins (peroxisome proliferator-activated receptor γ coactivator, PGC1α; and nuclear respirator factor 1, NRF1) and mitochondrial transcription factor A (TFAM) were increased by tDCS, suggesting that its biogenetic effect might be exerted at the level of transcription and replication of mitochondrial DNA [17]. Also, tDCS recovered an MTPT-induced increase of dynamin-related protein 1 (Drp1) expression, which reflects mitochondrial fragmentation and the release of pro-apoptotic proteins [50].

In PD, oxidative stress (OxS), mitochondrial dysfunction, excitotoxicity, and neuroinflammation are strictly linked to autophagy pathways. Autophagy is a cellular homeostatic process involved in both unspecific bulk degradation (macroautophagy, referred to simply as “autophagy”) of cytosolic proteins, aggregates and organelles [51] but also in specific catabolism (chaperone-mediated autophagy, CMA) of neuropathological proteins, including alpha-synuclein [52]. Defects in macroautophagy and CMA have been shown to play an important role in the pathogenesis of the disease [53]. To date, while no data are available in the literature on the specific effect of electrical stimulation on CMA, with the only exception of a recent study from our group [34], studies have investigated the effect on macroautophagy. Lee et al., 2018 [45] demonstrated that anodal tDCS over the left motor cortical area stabilizes the autophagy processes activated by MPTP-induced toxicity, as showed by microtubule-associated protein 1 light chain 3 (LC3) and AMP-activated protein kinase (AMPK) upregulation, and the mechanistic target of rapamycin (mTOR) and sequestosome1/p62 (p62) downregulation in experimental mice (see Table 1). However, such an effect has not been described for MTPT treatment-free, suggesting that the effect of anodal tDCS might occur under stress conditions. Also, anodal tDCS modulates the MPTP-induced upregulation of α-synuclein in substantia nigra pars compacta, which has been identified as a distinctive marker for PD [54]. As for the antioxidative effects, however, it is still under debate whether these effects on autophagy are a direct effect of tDCS, or rather a result of an increased release of BDNF [55].

Overall, these results represent a theoretical basis for the study of tDCS (anodal polarity) as a potential neuroprotective rather than a symptomatic therapy, as has mostly been considered so far. This application would be of great interest, since there is an absolute unmet need for treatments aiming to halt or restore the disease.

### 2.2. rTMS

A recent study evidenced a neuroprotective effect of early rTMS, as suggested by its ability to preserve tyrosine hydroxylase- (TH-) positive neurons in the substantia nigra pars compacta (SNpc) and fibers in the striatum in a hemiparkinsonian rat model induced by unilateral injection of 6-hydroxydopamine (6-OHDA) [56]. Furthermore, a previous study performed in parkinsonian rats induced by the inhibition of the ubiquitin-proteasome system, another catabolic pathway different from autophagy, whose dysfunction also exerts a pathogenic role in PD, demonstrated that rTMS exerts neuroprotective effects by alleviating the loss of TH-positive dopaminergic neurons, by preventing the loss of striatal dopamine levels, by reducing the levels of apoptotic protein (cleaved caspase-3) and inflammatory factors (cyclooxygenase-2 and tumor necrosis factor alpha) in lesioned substantia nigra [57].

## 3. NIBS and Neuroprotection in Alzheimer’s Disease

Alzheimer’s disease (AD) is a neurodegenerative disorder clinically characterized by amnestic and non-amnestic cognitive impairments. In pathological neurons, β-amyloid (Aβ)-containing extracellular plaques and tau-containing intracellular neurofibrillary tangles determine the aggregation of misfolded proteins, which leads to microtubule disorganization, cholinergic dysfunction, neuroinflammation, OxS, and, ultimately, neural dysfunction and synaptic loss [58]. Current pharmacological treatments for AD are mostly symptomatic, and therapies altering the underlying pathological processes are not commonly available. Similar to PD, non-invasive brain stimulation (NIBS) techniques (e.g., tDCS, transcranial magnetic stimulation—TMS) were shown to improve AD symptoms (e.g., global cognition, cognitive and memory functions, executive performance) [59,60] (see Figure 2). Several randomized clinical trials have been conducted by using either tDCS or rTMS for the treatment of cognitive symptoms associated to AD [61,62,63]. However, to date, the biochemical mechanisms are still not fully understood (see Table 2). Neurotrophic factors (NTFs) regulate the growth, survival, proliferation, migration and differentiation of neurons [64] and have been extensively studied in the context of AD. In AD, the lowered expression of NTFs, such as nerve growth factor (NGF) [65], BDNF [66], glial cell line-derived neurotrophic factor (GDNF) [67] and ciliary neurotrophic factor (CNTF), have been observed in affected brain regions, including the temporal cortex and hippocampus [68]. Recently, particular interest has been aroused by the potentially beneficial effect of neuromodulation techniques on BDNF, which is required in the hippocampus for late-phase long-term potentiation and represents one of the most important cellular mechanisms that underlies learning and memory (see Table 2). Moreover, BDNF induces the secretion of acetylcholine by enhancing the differentiation and survival of cholinergic neurons in the basal forebrain [69]. Notably, various studies have recently shown increasing BDNF levels in the basal forebrain and hippocampus in AD animal models [70] as well as in the serum of AD patients [71] after rTMS when compared to controls. Similarly, rTMS was found to be effective on NGF brain levels [72,73]. Moreover, rTMS and tDCS were effective also on the BDNF-TrkB signalling pathway [38,55,74], which affects cell survival, migration, outgrowth of axons and dendrites, synaptogenesis, synaptic transmission, and synapse remodelling [75].

Beyond enhancing neuron survival, rTMS and tDCS concurrently inhibit apoptosis [76,77]. Specifically, TMS effectively balanced the apoptotic pathways in an AD-mice model [72], by inhibiting pro-apoptotic members of the Bcl-2 family, such as cleaved caspase-3 and Bax, which are usually overexpressed in AD. Conversely, the same study pointed out TMS-induced reduced ubiquitination of beta catenin, which notoriously promotes cell survival.

Brain tissue in AD patients is characterized by increased oxidative stress (OxS), due to an imbalance in Reactive Oxygen Species (ROS) and Reactive Nitrogen Species (RNS) levels and the antioxidant defense system, resulting in damage to proteins, lipids, and DNA oxidation/glycoxidation processes [78]. Velioglu and co-workers pointed out beneficial effects of rTMS on oxidative stress levels in AD patients by applying rTMS over the lateral parietal cortex [71]. In AD, OxS contributes to endothelial Nitric Oxide (NO) depletion [79,80] and quickening cognitive decline [81] due to cerebral hypoperfusion. Conversely, cerebral hypoperfusion is responsible for increased OxS. Trivedi et al. [82] applied low electrical fields to endothelial cells to induce the increase of NO levels, and, in turn, vasodilatation. Similarly, Marceglia et al. [83] speculated that tDCS may raise the NO level to prevent brain and hippocampal hypoperfusion. Further studies investigated NIBS’s beneficial effect on brain perfusion via blood-brain barrier modulation. Neurons, glial cells and cerebral blood vessels are closely related in structure and function, collectively referred to as the “neurovascular unit” (NVU), which is critical for regulating cerebral blood flow, maintaining both the blood-brain barrier integrity and signal transduction between cells. In AD brains, there is evidence that the cerebral microcirculation system is damaged and the main components of NVU underwent pathological changes [84]. tDCS has been demonstrated to decrease the number of glial cells and increase levels of vascular endothelial growth factor (VEGF) and interleukin-8, possibly resulting in decreased local inflammation, increased vascularization, improved toxic metabolite clearance and microcirculation protection [85,86,87]. A further study [88] showed that tDCS-treated AD model mice exhibited reduced glial fibrillary acidic protein (GFAP) levels. GFAP plays a crucial role in endothelial junction function and morphologic changes of astrocyte end foot processes [89]. NIBS might also prevent Aβ 1-42 aggregation, increasing Aβ serum levels and this was assessed by both tDCS in AD patients [90] and tACS in AD-mouse model [91], presumably via microglia activation [92].

## 4. Beyond Neurodegenerative Disorders: Neuroprotection and Stroke

Cerebrovascular disorders are far from the scope of our paper. Moreover, recent studies about neuroprotection and NIBS in stroke have been conducted on animal models only [93]. These studies suggest that NIBS, especially tDCS, have different effects on ischemia, which can be defined as “neuroprotective”, i.e., by modifying motor recovery over time; in particular, in rat models, non-invasive stimulation reduces blood-brain barrier (BBB) disruption [94], promotes microglia polarization [95] and decreases spreading depolarization [96]. However, these results have been obtained using different approaches and whether these effects lead to a clinical benefit is still a matter of debate. Moreover, it is worth noting that these models are based on an experimentally induced ischemia, so it is difficult to compare them with what happens in human beings.

## 5. Differences and Similarities in Neuroprotective Mechanisms between NIBS and DBS

Deep brain stimulation (DBS) consists in implantation of electrodes into specific regions of the brain for chronic transmission of continuous electrical stimulation in high frequency from an implantable pulse generator [97]. This neurosurgical model has become a well-established therapy in advanced stages of PD, in essential tremor and dystonia, with immediate beneficial effects on the motor symptoms [98]. New applications of DBS are now emerging in the field of other neurodegenerative diseases like AD and other neuropsychiatric conditions [22]. Besides clinical benefit, DBS has shown to chronically induce a reorganization of neural activity, influencing synaptic plasticity, neurotransmission and neurogenesis [99]. Also, some findings suggest a possible neuroprotective role [14], although the underlying mechanisms are still unclear. In this chapter, we focus on the possible neuroprotective effect of the DBS in neurodegenerative disease (i.e., PD and AD), and we compare the results with those from NIBS studies (see Figure 3).

### 5.1. DBS and NIBS: Neuroprotection in PD

It has been proposed that chronic subthalamic nucleus DBS (STN-DBS) might attenuate disease progression by preserving nigral dopamine neurons from degeneration [100,109,110,111], as reported for tDCS [17,44]. Although some results do not support this hypothesis [112,113], in MPTP-treated monkeys, the implantation of STN-DBS demonstrated to protect dopaminergic neuronal loss in the substantia nigra pars compacta (SNpc) and periaqueductal grey matter (PAG) [100,109]. Also, studies on 6-OHDA-treated rats, evaluated 5–7 days and 14 days after the 6-OHDA administration, found a significant increase in preservation of dopaminergic nigral neurons on the lesioned side [110,111]. Similar results, but with a different animal model (i.e., MPTP-treated mice [17] and 6-OHDA-treated mice [44]), were reached with anodal tDCS. Similarly, in line with tDCS results [17,44], Spieles-Engmann et al., 2020 [111] found that STN-DBS provided significant sparing of DA neurons in the SN of rats two and four weeks after 6-OHDA administration, when 50% of the DA cell loss had already occurred. This effect was dependent upon proper electrode placement within the STN [111].

The neuroprotective potential of STN-DBS was also studied in PD rat models undergoing genetic overexpression of viral vector wild-type alpha-synuclein, but with poor results [114]. Indeed, STN-DBS did not protect the striatal denervation and SNpc neuronal loss [114]. tDCS, at the cellular level, increases the release of neurotrophic factors (e.g., BNDF) [45]. The same effect was found for STN-DBS, which raised the level of neurotrophic factors (e.g., BNDF) in the nigrostriatal system and primary motor cortex [102]. The signalling created by BDNF binding with tropomyosin-related kinase type B (trkB), that is believed to occur at dendrites, triggers three different intracellular cascades: phospholipase Cgamma/protein kinase C (PLCgamma/PKC), which promotes the regulation of synaptic plasticity; mitogen-activated protein kinase/extracellular signal related-kinase (MAPK/ERK), which is involved in the protein synthesis; phosphatidilnositol 3-kinase/protein kinase B (PI3K/Akt), which inhibits apoptosis and regulates translation/trafficking. Regarding activation of this molecular pathway, STN-DBS was demonstrated to induce Akt phosphorylation in a study using 6-OHDA rat PD model [103]. A large number of glutamatergic efferents from the STN to the dopaminergic neurons in the SNpc becomes overactive in the abnormal parkinsonian basal ganglia network and has recently been implicated as an aetiological factor in the progressive decline of intact dopamine neurones in the SNc [115]. Some studies support the theory that the mechanism at the basis of the neuroprotection of the SNpc in STN-DBS is the reduction of overreactive glutamatergic projections originating from the STN [104,116]. Another study has recently shown that in a PD rat model, using viral vector-mediated nigrostriatal overexpression of human A53T alpha-synuclein, STN-DBS rescued tyrosine-hydroxylase (TH) expression in SNpc neurons and improved motor fluctuations, with no effect on loss of striatal dopamine levels [101]. Anodal tDCS demonstrated a close biochemical effect in MPTP-treated mice [19,45].

### 5.2. DBS and NIBS: Neuroprotection in AD

Recent investigations have discovered that NIBS and DBS might share neuroprotective mechanisms. For example, mice studies have demonstrated that DBS in the nucleus basalis of Meynert (NBM-DBS) increases the cholinergic transmission in the neocortex, inducing high secretion of NGF [105,106,117]. Similar effects were shown for NIBS techniques [72,73]. NGF is a factor that has an important role in the function and survival of cholinergic basal forebrain neurons [118] and in the improvement of synaptic plasticity [119]. From a histochemical point of view, NBM-DBS down-regulated A-beta40 and A-beta42, increasing the survival of neurons and reducing apoptotic cells in the hippocampus and cortex [106]. Also, by stimulating residual neurons, NBM-DBS reduced the progression of cognitive decline in patients in the early stages of disease [105,117]. The reduction of Aβ 1-42 aggregation is another mechanism of neuroprotection proposed for anodal tDCS [88,90], but also found during DBS [107,120]. For example, in 2019, Leplus et al. [120] showed, in implanted transgenic AD rat-model (TgF344-AD), a reduction of neuronal loss and amyloid burden in the hippocampus and cortex and a decrease of microglial proliferation, responsible for the altered clearance of Aβ. Also DBS in Enthorinal Cortex (EC-DBS) demonstrated improvement in cognitive domain, as spatial and contextual memory, in wild-type rat model [107], with a reduction in amyloid plaque burden [107]. Akwa et al. [108], using triple transgenic mouse model of AD, showed that EC-DBS rescued synaptophysin levels, which is implicated in synaptic transmission, counteracting the negative effects of tau phosphorylation in AD. This mechanism may involve the autophagic-lysosomal clearance of pathological forms of tau protein [121].

In contrast to NIBS [83,88], only few data support the role of DBS in the modulation of NO levels and, consequently, brain vasodilatation [122,123]. Phase I and II studies in patients with AD, using the Forniceal-DBS, have revealed a significant improvement of the metabolism in temporoparietal region, paralleled by a reduction of hippocampal atrophy, with no clinical improvements at one year [122,123].

Interestingly, growing evidence supporting a disease-modifying action of DBS comes from psychiatric disorders. Recent studies in animal models of schizophrenia have proved that DBS of the medial pre-frontal cortex (mPFC) can prevent the enlargement of lateral ventricle volumes, a marker of disease severity and progression, as well as the mal-development of both serotoninergic and glutamatergic transmission [124]. However, no corresponding considerations can be made for NIBS techniques.

## 6. Limitations and Future Directions

Despite the safety of these techniques, as demonstrated during years of clinical applications, studies regarding NIBS and neuroprotection have some limitations. First, in vitro and animal models do not completely fit with the complexity of human behavior. Also pathological bases may differ [125,126]; although animal models frequently express specific genes engaged in human neurodegenerative disorders, it is still debated whether there are intracellular aggregates of pathological proteins driving the disease course over time [127,128]. Another limitation is the deep localization of nuclei involved in the early phase of neurodegeneration; current modelling studies have explored the distribution of electric fields through the scalp, but little is known about the possibility to interfere with deepest structures, including basal ganglia and thalamus. An innovative approach has been recently provided by the use of the so-called non-invasive deep brain stimulation (NDBS), which is based on the collimation of different electric fields able to modulate deep nuclei in the brain and brainstem [129,130]; this method is safe, non-invasive and aims to assess if group-level hotspots exist in deep brain areas by using different electrode montages and has been successfully applied for the treatment of refractory temporal lobe epilepsies [131].

Finally, the same neurological disorder can be characterized by different pathological proteins, and, in some cases, pathological mechanisms are only poorly understood.

## 7. Conclusions

Neuroprotective and disease-modifying effects underlying the action of either invasive and non-invasive brain stimulating techniques represent a novel, promising and undervalued field of research. Recent evidence supports the role of tDCS in the modulation of the aggregation state and accumulation of pathological proteins. Nonetheless, only few studies have been recently developed using rTMS, whereas the literature describing a clear disease-modifying effect of DBS is scarce. DBS has recently shown neuroprotective effects in non-degenerative disorders, especially in psychiatric illnesses. tDCS seems to interfere both with lysosomal and non-lysosomal pathways, with the polarity of after-effects strictly depending on the aggregation state of pathological proteins and the pre-existing excitability state [34,132]. In this scenario, the classical, and highly debated, view supporting polarity-dependent after-effects, with cathodal tDCS inducing inhibition and anodal polarization leading to excitation [133], can be overcome. A third mechanism underlying a neuroprotective effect refers to the modulation of the inflammatory response, also by interfering with BBB permeability. This mechanism is also corroborated by findings in animal models of stroke [94]. Although α-synuclein and β-amyloid differ in terms of the number of aggregation states and intra-versus extra-cellular localization, encouraging results are coming from animal and cellular models of both PD and AD. Future studies should be devoted to evaluating the impact of NIBS in other “proteinopathies”, including Fronto-Temporal Dementias (FTDs) and Amyotrophic Lateral Sclerosis (ALS). Moreover, the neuroprotective role of so-called Non-Invasive Deep Brain Stimulation (NDBS) should be explored. NDBS refers to techniques currently under investigation that would allow to target deep brain regions through NIBS techniques [134]. Since tDCS and tACS are able to affect deep substrates primarily implicated in neurological pathologies (e.g., subthalamic nucleus for PD) [6], their neuroprotective action might be directed towards those structures, with huge clinical impact. However, no standardized and recognized stimulation protocols are now available for selective and precise targeting.

## Figures and Tables

**Figure 1 ijms-23-13775-f001:**
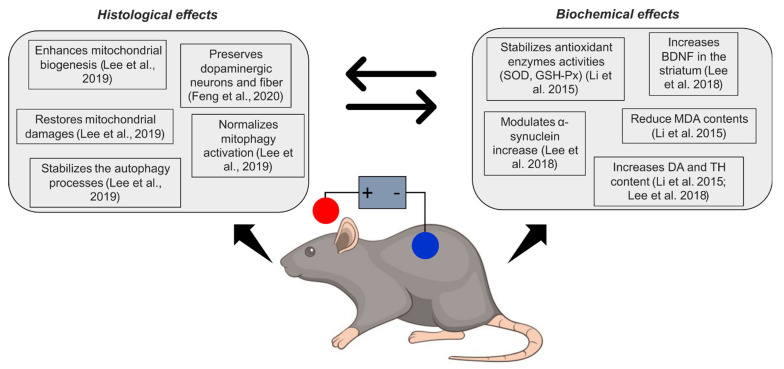
Schematic overview of neuroprotective effects of tDCS in animal models of Parkinson’s disease. SOD = superoxide dismutase; GSH-Px = glutathione peroxidase; BDNF = brain-derived neurotrophic factor; MDA = malonaldehyde; DA = dopamine; TH = tyrosine hydroxylase. Feng et al., 2020 [44]; Lee et al., 2019 [17]; Li et al., 2015 [19]; Lee et al., 2018 [45].

**Figure 2 ijms-23-13775-f002:**
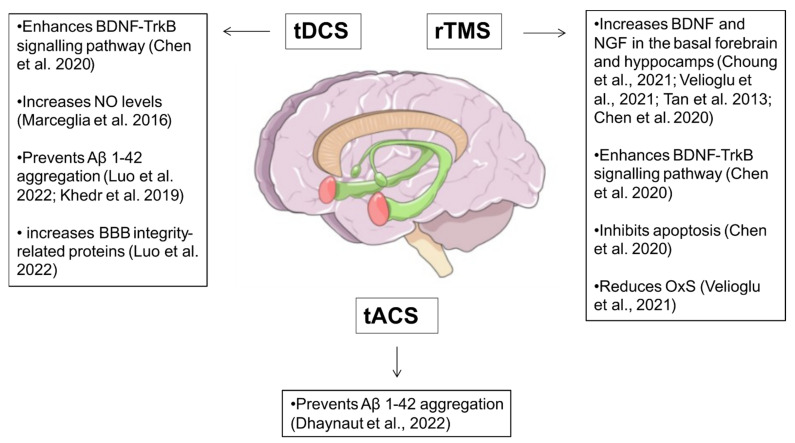
Schematic overview of neuroprotective effects of NIBS in Alzheimer’s disease. tDCS = transcranial direct current stimulation; tACS = transcranial alternating current stimulation; rTMS = repetitive transcranial magnetic stimulation; BDNF = brain-derived neurotrophic factor; TrkB = Tropomyosin receptor kinase B; NGF = Nerve growth factor; NO = nitric oxide; OxS = oxidative stress. Choung et al., 2021 [70]; Velioglu et al., 2021 [71]; Tan et al., 2013 [73]; Chen et al., 2020 [74]; Marceglia et al., 2016 [83]; Luo et al., 2022 [88]; Khedr et al., 2019 [90].

**Figure 3 ijms-23-13775-f003:**
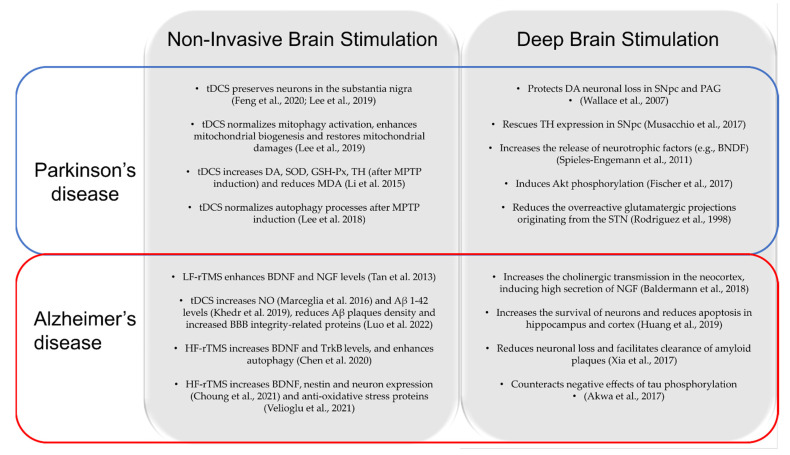
Schematic overview of comparisons in neuroprotective effects between NIBS and DBS. tDCS = transcranial direct current stimulation; DA = dopamine; SOD = superoxide dismutase; GSH-PX = glutathione peroxidase; TH = tyrosine hydroxylase; MPTP = 1-methyl-4-phenyl-1,2,3,6-tetrahydropyridine; MDA = malonaldehyde; LF-rTMS = low frequency repetitive transcranial magnetic stimulation; BDNF = Brain-Derived Neurotrophic Factor; NGF = nerve growth factor; NO = nitric oxide; Aβ = amyloid-β peptide; BBB = blood brain barrier; HF-rTMS = high frequency repetitive transcranial magnetic stimulation; TrkB = Tropomyosin receptor kinase B; SNpc = substantia nigra pars compacta; PAG = periaqueductal grey matter; Akt = Protein kinase B; STN = subthalamic nucleus. Feng et al., 2020 [44]; Lee et al., 2019 [17]; Li et al., 2015 [19]; Lee et al., 2018 [45]; Tan et al., 2013 [73]; Marceglia et al., 2016 [83]; Khedr et al., 2019 [90]; Luo et al., 2022 [88]; Chen et al., 2020 [74]; Choung et al., 2021 [70]; Velioglu et al., 2021 [71]; Wallace et al., 2007 [100]; Musacchio et al., 2017 [101]; Spieles-Engemann et al., 2011 [102]; Fischer et al., 2017 [103]; Rodrigues et al., 1998 [104]; Baldermann et al., 2018 [105]; Huang et al., 2019 [106]; Xia et al., 2017 [107]; Akwa et al., 2017 [108].

**Table 1 ijms-23-13775-t001:** Studies assessing neuroprotective effects of tDCS in animal models of Parkinson’s disease.

Study	Sample/Animals	Polarity	Configuration	Parameters	Biological Outcomes	Biological Results
Li et al., 2015 [19]	36 C57Bl mice (*n* = 9 in control group; *n* = 9 in sham tDCS group; *n* = 9 in tDCS groups; *n* = 9 in drug group)	Anodal/Sham	AE: left frontal cortex; R: between the shoulders	0.2 mA, 10 min/day, 21 consecutive days AEA: 3.5 mm^2^ CD: 5.7 mA/cm^2^	DA, TH, SOD and GSH-PX activities, nonenzymatic MDA activity	tDCS increased DA, SOD and GSH-Px; after MPTP induction, anodal tDCS increased TH and reduced MDA
Lee et al., 2018 [45]	60 Male C57BL/6 mice (*n* = 15 in control group; *n* = 15 in anodal tDCS group; *n* = 15 in MPTP group; *n* = 15 in MPTP + tDCS group)	Anodal/Sham	AE: left motor cortex; R: between the shoulders	0.1 mA, 30 min/day, 5 consecutive days AEA: 3.1 mm^2^ CD: 3.2 mA/cm^2^	TH-positive cells; TH; α-synuclein protein; loss of dopaminergic neuron cells; ratio of LC3-II/LC3-I; p62; PI3K; mTOR; AMPK; ULK	tDCS attenuated decrease of TH, p62, mTOR, PI3K, BDNF; attenuated increase of α-synuclein, LC3-II/LC3-I, AMPK and ULK
Lee et al., 2019 [17]	Male C57BL/6 mice (number n.r.)	Anodal/Sham	AE: on motor cortex; R: between the shoulders	0.1 mA, 30 min/day, 5 days/week, 1 week; AEA: 3.1 mm^2^ CD: 3.2 mA/cm^2^	Expression of: TH, mitophagy-related proteins; marker of degradation phase of autophagy; mitochondrial biogenesis-related proteins; mitochondrial fission and fusion -related proteins; ATP concentration. Mitochondrial GDH activity	tDCS preserved neurons and fibers in substantia nigra and striatum; attenuated mitochondrial GDH activity, ATP concentration; increased mitophagy-related and mitochondrial biogenesis proteins
Feng et al., 2020 [44]	16 male Wistar (*n* = 8 in anodal group; *n* = 8 in sham group)	Anodal/Sham	AE: skull bregma; R: anterior chest	300 μA, 20 min/day, 5 days/week, 4 weeks; AEA: 37.9 mm^2^; CD: 0.16 mA/cm^2^	Loss of dopaminergic nigrostriatal neurons and fibers	tDCS preserved neurons in the substantia nigra, but not fibers in the striatum

tDCS = transcranial direct current stimulation; AE = active electrode; R = reference; AEA = active electrode area; CD = current density; DA = dopamine; TH = tyrosine hydroxylase; SOD = superoxide dismutase; GSH-PX = glutathione peroxidase; MDA = malonaldehyde; MPTP = 1-methyl-4-phenyl-1,2,3,6-tetrahydropyridine; LC3 = microtubule-associated protein 1 light chain 3; p62 = sequestosome 1/p62; PI3K = phosphoinositide 3-kinases; mTOR = mechanistic target of rapamycin; AMPK = AMP-activated protein kinase; ULK = unc-51-like kinase 1; ATP = adenosine triphosphate; GDH = glutamate dehydrogenase.

**Table 2 ijms-23-13775-t002:** Studies assessing neuroprotective effects of NIBS in animal models of Alzheimer’s Disease (AD).

Study	NIBS Method	Sample/Animals	Configuration	Parameters	Biological Outcomes	Biological Results
Tan et al., 2013 [73]	rTMS (LF)	84 mice (*n* = 21 in control group; *n* = 21 rTMS group; *n* = 21 in Aβ injection; *n* = 21 Aβ injection + rTMS)	Whole brain stimulation	400 pulses per session, 7 days/week, 2 weeks LF-rTMS: 20 trains (20 pulses at 1 Hz, 10 s inter-interval)	Neuroplasticity-related proteins (BDNF, NGF and NMDA receptor) levels	LF-rTMS reversed NMDA receptor suppression, enhanced, BDNF and NGF levels
Marceglia et al., 2016 [83]	tDCS (anodal/sham)	7 AD patients (*n* = 7 tDCS; *n* = 7 sham)	AE: bilateral temporo-parietal area; R: right arm	1.5 mA, 15 min/day, 1 day AEA: 25 cm^2^ CD: 0.06 mA/cm^2^	total NO levels	tDCS increased NO levels
Khedr et al., 2019 [90]	tDCS (anodal)	46 AD patients (*n* = 23 tDCS; *n* = 23 sham)	AE: bilateral temporo-parietal area; R: left arm	2 mA, 20 min each side (5 min in between), 5 days/week, 2 weeks AEA: 35 cm^2^ CD = 0.057 mA/cm^2^	AD brain damage biomarkers levels (TAU and Aβ 1-42)	tDCS increased Aβ 1-42
Chen et al., 2020 [74]	rTMS (HF)	30 mice (*n* = 15 rTMS; *n* = 15 sham)	Whole brain stimulation,	600 pulses per session, 7 days/week, 2 weeks HF-rTMS 20 trains (30 pulses at 5 Hz, 2 s inter-interval)	Synaptic plasticity-related proteins (PSD95), neurotrophic factors (BDNF, TrkB and AKT), autophagy marker proteins (p62 and LC3-II/LC3-I)	HF-rTMS increased BDNF and TrkB levels, and enhanced hippocampal cellular autophagy
Choung et al., 2021 [70]	rTMS (HF/LF/sham)	24 mice (*n* = 8 HF-rTMS; *n* = 8 LF-rTMS; *n* = 8 sham)	Whole brain stimulation	1600 pulses per session, 5 days/week, 2 weeks HF-rTMS: 40 trains (2 s duration at 20 Hz, 28 s inter-interval) LF-rTMS: continuous stimulation (1 Hz).	BDNF, nestin and neuron protein levels	HF-rTMS increased BDNF, nestin and neuron expression levels in hippocampus and cortex, compared to sham
Velioglu et al., 2021 [71]	rTMS (HF)	15 subjects	Left parietal cortex stimulation	1640 pulses per session, 5 days/week, 2 weeks HF-rTMS: 42 trains (2 s duration at 20 Hz, 28 s inter-interval)	BDNF and anti-oxidative stress proteins levels	HF-rTMS increased BDNF and anti-oxidative stress proteins levels
Luo et al., 2022 [88]	tDCS (anodal/sham)	33 AD model mice (*n* = 11 tDCS; *n* = 11 not treated; *n* = 11 sham)	AE: frontal cortex; R: thorax	150 µA, 30 min/day, 5 days/week, 2 weeks AEA:nr CD: nr	Aβ plaques density in the hippocampus and frontal cortex, NVU integrity	tDCS reduced Aβ plaques density and increased BBB integrity-related proteins

LF-rTMS = low frequency repetitive transcranial magnetic stimulation; Aβ = amyloid-β peptide; BDNF = Brain-Derived Neurotrophic Factor; NGF = Nerve growth factor; NMDA = N-methyl-D-aspartate receptor; tDCS = transcranial direct current stimulation; AD = Alzheimer’s disease; AE = active electrode; R = reference; AEA = active electrode area; CD = current density; NO = nitric oxide; TAU = tubulin associated unit; HF-rTMS = high frequency repetitive transcranial magnetic stimulation; PSD95 = Postsynaptic density protein 95; TrkB = Tropomyosin receptor kinase B; AKT = protein kinase B; LC3 = microtubule-associated protein 1 light chain 3; p62 = sequestosome 1/p62; NVU = neurovascular unit; BBB = blood brain barrier.

## Data Availability

Not applicable.

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
