# Peer review of "Neuroprotection and Non-Invasive Brain Stimulation: Facts or Fiction?"

_ijms, 2022, doi:10.3390/ijms232213775_

Round 1

Reviewer 1 Report

The article ”Neuroprotection and Non-Invasive Brain Stimulation: facts or fiction?” can be considered a good one in the context of actual data on Non-Invasive Brain Stimulation techniques and neurodegenerative diseases. Even so, the reviewer's job is to underline the weak points of the manuscript in order to determine a further improvement of it. Considering this task, in my view the main points of improvement are the following:

1. The title of the article suggests that will be a presentation of facts and fiction about Non-Invasive Brain Stimulation techniques. There are many facts presented, but there are no underlined or discussed ”fiction” data about the same techniques. Which are their limits and also why these techniques are promoted as future treatments?

2. The title of the article suggests rather a systematic review rather than a narrative one, so my second question is why the authors have chosen this kind of article and not a systematic one, given the fact that the title suggests it and many articles on this subject have been published also in the last three years for example.

3. Why the pathology was limited to Parkinson's Disease and Alzheimer's Disease, and why not Stroke or other pathologies in which the same techniques can be used? Are there data for limiting these techniques to the chosen neurodegenerative diseases?

4. The references used are very supportive of the manuscript, but why not more articles from the last at least two years ?, given the title, which can also be found on MDPI with a similar subject: Non-Invasive interventions in neurodegenerative diseases.

5. The discussion is the weak point of the article, as it is included in other subpoints of the manuscript.

6. The concept of neuroprotection can be more precisely presented in order to avoid the fall between fact and fiction. Why does it happen to be such a debate between fact and fiction in the field? What triggers this kind of discussion?

7. Are there bad results or side effects when using Non-Invasive Brain Stimulation techniques to be presented?

8. Figure 1 can be improved with larger boxes and some connections between various effects maybe.

9. Are clinical trials available for Non-Invasive interventions in neurodegenerative diseases? 

10.  Can be explained clearly the difference between anodal and cathodal interventions of transcranial Direct Current Stimulation and how this influence the outcomes?

Overall, I consider this manuscript of a high scientific level, and the above queries are meant to be constructive and helpful in improving it.

Author Response

Response to Reviewer 1 Comments

The article ”Neuroprotection and Non-Invasive Brain Stimulation: facts or fiction?” can be considered a good one in the context of actual data on Non-Invasive Brain Stimulation techniques and neurodegenerative diseases. Even so, the reviewer's job is to underline the weak points of the manuscript in order to determine a further improvement of it. Considering this task, in my view the main points of improvement are the following:

Response: We thank the reviewer for the comment. We have amended the manuscript following the suggestions. We hope the manuscript might be suitable for a publication in its reviewed form.

  1. The title of the article suggests that will be a presentation of facts and fiction about Non-Invasive Brain Stimulation techniques. There are many facts presented, but there are no underlined or discussed ”fiction” data about the same techniques. Which are their limits and also why these techniques are promoted as future treatments?

Response 1: We thank the reviewer for the comment. We have added a specific section (paragraph 5, page 15, line 357-373) to disucss the “fictions”, meaning the current limitiations and future directions. In particular, in vitro and animal models do not completely fit with the complexity of human behavior. Although animal models frequently express specific genes engaged in human neurodegenerative disorders, it’s still debated whether there are intracellular aggregates of pathological proteins driving disease course over time. Another limitation is the deep localization of nuclei involved in the early phase of neurodegeneration; current modelling studies have explored the distribution of electric fields through the scalp, but little is known about the possibility to interfere with deepest structures, including basal ganglia and thalamus.

  1. The title of the article suggests rather a systematic review rather than a narrative one, so my second question is why the authors have chosen this kind of article and not a systematic one, given the fact that the title suggests it and many articles on this subject have been published also in the last three years for example.

Response 2: We thank the reviewer for the comment. We chose to write a manuscript in the form of “narrative review” because, although the results, and more in general, the knowledge on neuroprotective effects of NIBS is currently growing, still the data available are too few for a systematic review. Also, the form of “narrative review” allows the authors to be more inclusive and descriptive in terms of concepts to be discussed.

  1. Why the pathology was limited to Parkinson's Disease and Alzheimer's Disease, and why not Stroke or other pathologies in which the same techniques can be used? Are there data for limiting these techniques to the chosen neurodegenerative diseases?

Response 3: We thank the reviewer for the comment. We decided to focus our manuscript on neurodegenerative diseases (PD and AD) because the available knowledge about neuroprotection of NIBS is mostly related to these two pathology. However, we added a new paragraph (paragraph 4, page 13, line 253-263) regarding the few results on stroke. In particular, in animal models, non-invasive stimulation is likely to reduce blood-brain barrier (BBB) disruption, promote microglia polarization and decrease spreading depolarization. Nonetheless, these results have been obtained using different approaches and whether these effects lead to a clinical benefit is still a matter of debate (paragraph 4, page 13, line 253-263).

  1. The references used are very supportive of the manuscript, but why not more articles from the last at least two years, given the title, which can also be found on MDPI with a similar subject: Non-Invasive interventions in neurodegenerative diseases.

Response 4: We thank the reviewer for the comment. In the revised version of our manuscript, the bibliography has been updated accordingly. In particular, most recent papers have been cited in new sections added, especially regarding stroke and DBS in psychiatric diseases.

  1. The discussion is the weak point of the article, as it is included in other sub-points of the manuscript.

Response 5: We thank the reviewer for the comment. This section has now been improved by adding a paragraph about possible limitations and fictions for the use of NIBS, as well as by discussing new concepts underlying the polarity-dependent after-effects of tDCS (page 16, lines 413-419). Finally, the section about NIBS, DBS and AD has been improved.  

  1. The concept of neuroprotection can be more precisely presented in order to avoid the fall between fact and fiction. Why does it happen to be such a debate between fact and fiction in the field? What triggers this kind of discussion?

Response 6: We thank the reviewer for the comment. We better specified this key point in the Discussion section, along with (page , lines ).

  1. Are there bad results or side effects when using Non-Invasive Brain Stimulation techniques to be presented?

Response 7: We thank the reviewer for the comment. We specified both in Introduction and Discussion sections (page 1, lines 41-42; page 16, lines 399-400) that these techniques are safe, adding related literature. A more detailed description of their side-effects are out of the scope of this review. 

  1. Figure 1 can be improved with larger boxes and some connections between various effects maybe.

Response 8: We thank the reviewer for the comment. We have amended the figure as suggested.

  1. Are clinical trials available for Non-Invasive interventions in neurodegenerative diseases? 

Response 9: We thank the reviewer for the comment. Unfortunately, no clinical trials assessing neurprotective effect of NIBS are available for PD, while the three studies cited in the manuscript (Velioglu et al., 2021; Marceglia et al. 2016; Khedr et al. 2019) are available for AD.

  1. Can be explained clearly the difference between anodal and cathodal interventions of transcranial Direct Current Stimulation and how this influence the outcomes?

Response 10: We thank the reviewer for the comment. We have better discussed this point at page (lines ). In particular, tDCS seems to interfere both with lysosomal and non-lysosomal pathways, with the polarity of after-effects strictly depending on the aggregation state of pathological proteins and the pre-existing excitability state. In this scenario, we suggest that the classical, view supporting polarity-dependent after-effects, with cathodal tDCS inducing inhibition and anodal polarization leading to excitation, should be overcome (page 16, lines 413-421).

Overall, I consider this manuscript of a high scientific level, and the above queries are meant to be constructive and helpful in improving it.

Response: Once again, we thank the reviewer for the comment. We hope the manuscript might be suitable for a publication in its reviewed form.

Reviewer 2 Report

In this narrative review, Guidetti et al. attempt to summarize evidence for a neuroprotective effect of non-invasive brain stimulation (NIBS) in neurodegenerative disorders Alzheimers’ (AD) and Parkinson’s Disease (PD). Although the topic is interesting and clinically meaningful, I have the following concerns:

Major points:

1.       Although the title suggests a focus on NIBS, section 4 of this manuscript is about the comparison between NIBS and deep brain stimulation (DBS).

2.       Actually, section 4 is mostly a summary of DBS studies and not a comparison between DBS and NIBS. Reconsider what kind of information you want to summarize in this review and then change the title or the content

3.       Related to the former point, section 4 would benefit from a chart or table that schematically compares NIBS and DBS

4.       The manuscript is about NIBS and DBS in neurodegenerative disorders. This is interesting, given that a lot of research on NIBS, especially on therapeutic rTMS, is focused on psychiatric disorders. It is therefore surprising and rather misplaced that authors mention psychiatric disorders only in the context of DBS (line 58-60).

5.       The title of section 2.1 mentions ‘tDCS and tACS’. However, the section itself only summarizes studies on tDCS.

6.       The content is unbalanced. For example, 2.2 is much shorter compared to 2.1. Many studies on rTMS in PD are missing in 2.2.

7.       Consider structuring each individual section to make it easier for the reader to grasp its content. For example, 2.1 includes a summary sentence at the end, but this is not the case for other sections.

8.       There is a mismatch between the table content and the manuscript. For example, table 1 includes behavioral results. However, no such content can be found in the main text. Consider summarizing behavioral results in your manuscript as well.

Minor points:

1.       Only introduce an abbreviation (e.g., rTMS) once, there is no need to introduce it again in another section

2.       Transcranial electrical stimulation (TES) includes tDCS, tACS, tRNS. Consider using “TES” in the first sentence of the introduction instead of tDCS and tACS.

3.       The manuscript contains many grammatical and English errors.

4.       On page 7, the authors state that current pharmacological treatments for AD are only symptomatic. This is not correct, see for example aducanumab.

5.       Add references in the contents of both figures

6.       Change the title of Table 2. The table is not only focused on tDCS but also contains rTMS studies

Author Response

Response to Reviewer 2 Comments

In this narrative review, Guidetti et al. attempt to summarize evidence for a neuroprotective effect of non-invasive brain stimulation (NIBS) in neurodegenerative disorders Alzheimers’ (AD) and Parkinson’s Disease (PD). Although the topic is interesting and clinically meaningful, I have the following concerns:

Response: We thank the reviewer for the comment. We have amended the manuscript following the suggestions. We hope the manuscript might be suitable for a publication in its reviewed form.

Major points:

  1. Although the title suggests a focus on NIBS, section 4 of this manuscript is about the comparison between NIBS and deep brain stimulation (DBS).

Response 1: We thank the reviewer for the comment. We decided to include also DBS-induced neuroprotective effects in AD and PD as a matter of debate, in the light of the growing scientific interest in neurostimulation and neuroprotection. We modified the manuscript, clearly stating this aspect (page 1, line 27-31; page 2 line 56-57)

  1. Actually, section 4 is mostly a summary of DBS studies and not a comparison between DBS and NIBS. Reconsider what kind of information you want to summarize in this review and then change the title or the content

Response 2: We thank the reviewer for the comment. We have amended the section 4 (now section 5) to better underline the comparisons between DBS and NIBS (page 13, paragraph 5).

  1. Related to the former point, section 4 would benefit from a chart or table that schematically compares NIBS and DBS

Response 3: We thank the reviewer for the comment. We have added Figure 3 following this suggestion (page 14, with figure legend at lines 291-305).

  1. The manuscript is about NIBS and DBS in neurodegenerative disorders. This is interesting, given that a lot of research on NIBS, especially on therapeutic rTMS, is focused on psychiatric disorders. It is therefore surprising and rather misplaced that authors mention psychiatric disorders only in the context of DBS (line 58-60).

Response 4: We thank the reviewer for the comment. We have added a more detailed discussion about psychiatric illness both in Discussion and Conclusion sections.

  1. The title of section 2.1 mentions ‘tDCS and tACS’. However, the section itself only summarizes studies on tDCS.

Response 5: We thank the reviewer for the comment. We have corrected the title of the paragraph (page 2, line 63).

  1. The content is unbalanced. For example, 2.2 is much shorter compared to 2.1. Many studies on rTMS in PD are missing in 2.2.

Response 6: We thank the reviewer for the comment. In the revised version of our manuscript, we have tried to balance this ratio. Also Figure 3 contains and discusses many aspects of either tDCS or rTMS after-effects when compared to invasive brain stimulation techniques, such as DBS.

  1. Consider structuring each individual section to make it easier for the reader to grasp its content. For example, 2.1 includes a summary sentence at the end, but this is not the case for other sections.

Response 7: We thank the reviewer for the comment. Accordingly, we have re-structured the entire paper, also by adding a paragraph discussing limitations and fictions (page 16, lines 386-405), as well as by improving Conclusion section (page 16, lines 415-421).

  1. There is a mismatch between the table content and the manuscript. For example, table 1 includes behavioral results. However, no such content can be found in the main text. Consider summarizing behavioral results in your manuscript as well.

Response 8: We thank the reviewer for the comment. Given the topic of the review (biochemical more than behavioural), we decided to delate the behavioral results in the table to better underline the biological ones (Table I, page 5; Table II, page 10).

Minor points:

  1. Only introduce an abbreviation (e.g., rTMS) once, there is no need to introduce it again in another section

Response 9: We thank the reviewer for the comment. We have corrected this error throughout the whole article.

  1. Transcranial electrical stimulation (TES) includes tDCS, tACS, tRNS. Consider using “TES” in the first sentence of the introduction instead of tDCS and tACS.

Response 10: We thank the reviewer for the comment. We opted for introducing in the first lines these two techniques (tDCS and tACS) because they are the more used and known, while tRNS is more new and less studied. Also, neuroprotective evidence is currently available only for tDCS and tACS, in the context of tES techniques.

  1. The manuscript contains many grammatical and English errors.

Response 11: We thank the reviewer for the comment. The revised version of our manuscript has been extensively revised by a native.

  1. On page 7, the authors state that current pharmacological treatments for AD are only symptomatic. This is not correct, see for example aducanumab.

Response 12: We thank the reviewer for the comment. We have amended introducing the terms “Current pharmacological treatments for AD are mostly symptomatic, and therapies alter-ing the underlying pathological processes are not commonly available” (page 7, line 182-184)

  1. Add references in the contents of both figures

Response 13: We thank the reviewer for the comment. We have amended both the figures (page 4 and 8)

  1. Change the title of Table 2. The table is not only focused on tDCS but also contains rTMS studies

Response 14: We thank the reviewer for the comment. We have corrected this mistake.

Overall, I consider this manuscript of a high scientific level, and the above queries are meant to be constructive and helpful in improving it.

Response: Once again, we thank the reviewer for the comment. We hope the manuscript might be suitable for a publication in its reviewed form.

Round 2

Reviewer 1 Report

Dear authors, the manuscript in the revised version is properly improved and all my concerns and comments were adequately addressed, so I do not have further comments and conclude for acceptance for publication in the present form.